# COVID-19 Attitudes and Vaccine Hesitancy among an Agricultural Community in Southwest Guatemala: A Cross-Sectional Survey

**DOI:** 10.3390/vaccines11061059

**Published:** 2023-06-02

**Authors:** Neudy Rojop, Diva M. Calvimontes, Edgar Barrios, Molly M. Lamb, Alejandra Paniagua-Avila, Jose Monzon, Lindsey M. Duca, Chelsea Iwamoto, Anna N. Chard, Melissa Gomez, Kareen Arias, Yannik Roell, Guillermo Antonio Bolanos, Emily Zielinski-Gutierrez, Eduardo Azziz-Baumgartner, Maria Renee Lopez, Celia Cordon-Rosales, Edwin J. Asturias, Daniel Olson

**Affiliations:** 1Center for Human Development, Fundación para la Salud Integral de los Guatemaltecos, Retalhuleu 11010, Guatemala; rojophernandez@gmail.com (N.R.); diva.barrientos@cuanschutz.edu (D.M.C.); ebarrios1908@gmail.com (E.B.); melissa.gomez.fsigcu@gmail.com (M.G.); kareen.arias@gmail.com (K.A.); antonio.bolanos@cuanschutz.edu (G.A.B.); edwin.asturias@cuanschutz.edu (E.J.A.); 2Center for Global Health, Colorado School of Public Health, Aurora, CO 80045, USA; molly.lamb@cuanschutz.edu (M.M.L.);; 3Department of Epidemiology, Colorado School of Public Health, Aurora, CO 80045, USA; 4Department of Epidemiology, Mailman School of Public Health, Columbia University, New York, NY 11032, USA; mp3856@cumc.columbia.edu; 5Centers for Disease Control and Prevention (CDC), Atlanta, GA 30333, USAebz0@cdc.gov (E.Z.-G.); eha9@cdc.gov (E.A.-B.); 6Centro de Estudios en Salud, Universidad del Valle de Guatemala, Guatemala City 01015, Guatemala; mlopez@ces.uvg.edu.gt (M.R.L.); ccordon@ces.uvg.edu.gt (C.C.-R.); 7Department of Pediatrics, University of Colorado School of Medicine, Aurora, CO 80045, USA

**Keywords:** COVID-19, SARS-CoV-2, hesitancy, access, Guatemala, agricultural worker

## Abstract

Despite offering free-of-charge COVID-19 vaccines starting July 2021, Guatemala has one of the lowest vaccination rates in Latin America. From 28 September 2021 to 11 April 2022, we conducted a cross-sectional survey of community members, adapting a CDC questionnaire to evaluate COVID-19 vaccine access and hesitancy. Of 233 participants ≥ 12 years, 127 (55%) received ≥1 dose of COVID-19 and 4 (2%) reported prior COVID-19 illness. Persons ≥ 12 years old who were unvaccinated (*n* = 106) were more likely to be female (73% vs. 41%, *p* < 0.001) and homemakers (69% vs. 24%, *p* < 0.01) compared with vaccinated participants (*n* = 127). Among those ≥18 years, the main reported motivation for vaccination among vaccinated participants was to protect the health of family/friends (101/117, 86%); on the other hand, 40 (55%) unvaccinated persons reported little/no confidence in public health institutions recommending COVID-19 vaccination. Community- and/or home-based vaccination programs, including vaccination of families through the workplace, may better reach female homemakers and reduce inequities and hesitancy.

## 1. Introduction

Despite a significant disease and economic burden of COVID-19 and free-of-charge vaccination, low- and middle-income countries (LMICs) have struggled to achieve high vaccination rates. Limited access to vaccines and, increasingly, vaccine hesitancy fuel repeated waves of community transmission and waste valuable resources [1,2].

“Vaccine access” and “vaccine hesitancy” are two distinct but overlapping phenomena, which have both been identified as barriers to global vaccination in general and COVID-19 vaccination in particular [3,4]. Many populations in both high-income countries (HICs) and LMICs face ongoing barriers to vaccine access, such as decreased vaccine allocation and poorer public health and logistical infrastructure; these barriers were exacerbated by the COVID-19 pandemic [3,5,6].

In addition, both HICs with greater access to effective vaccines and, increasingly, LMICs have been impacted by vaccine hesitancy [4,7]. Vaccine hesitancy is defined by the World Health Organization (WHO) Strategic Advisory Group of Experts (SAGE) Working Group on Vaccine Hesitancy as “a delay in acceptance or refusal of vaccination despite availability of vaccination services” [8]. Importantly, they point out that vaccine hesitancy is “complex and context specific,” “varying across time, place and vaccines,” and “influenced by complacency, convenience and confidence.” Thus, addressing vaccine access and hesitancy requires a dynamic understanding of the local context.

Guatemala is one such country with a history of inconsistent vaccine access and high vaccine acceptance, though vaccine hesitancy in certain populations has been increasing, especially during the COVID-19 pandemic [9,10,11,12,13]. Guatemala’s health system is composed of two sectors, public and private. The former comprises the Ministry of Public Health and Social Assistance (MSPAS), which covers 70% of the population, including its immunization program; the Guatemalan Social Security Institute (IGSS), which provides coverage to 18% of the population linked to formal employment; and the Military Health Service, which provides health services to members of the armed forces and the police (less than 0.5%). The private sector includes civil society and/or religious organizations that operate on a not-for-profit basis, in addition to various for-profit providers. MSPAS is financed with resources from state tax revenues, aid, loans, and international donations.

Guatemala began its COVID-19 vaccination program on 25 February 2021, targeting health workers and individuals with comorbidities; adolescents aged 12–17 years were included as of 22 September 2021. Despite the increasing availability of free-of-charge COVID-19 vaccines, coverage in Guatemala remains among the lowest in Latin America, especially in rural and indigenous communities. As of 1 July 2022, only 46% of the eligible population in Guatemala had received one vaccine dose, and 35% had received two doses [14,15]. By April 2022, 1.47 million doses of U.S.-donated COVID-19 vaccine [16] and nearly 5 million doses of purchased Sputnik V vaccine [17] had expired, presumably because of low accessibility and demand throughout the country. 

Though limited data exist on the changing prevalence of vaccine hesitancy and access throughout Guatemala, especially in the setting of the COVID-19 pandemic, some populations have not been well characterized. One critical population is essential agricultural workers, who comprise 35% of Guatemala’s overall labor force and 11.3% of its gross domestic product. [18] As in many LMICs, Guatemalan agricultural workers not only play a critical role in the local food supply, but also an important role in global food security [19]. They are often the primary income earners for their households and, as essential workers, continued to work throughout the pandemic despite high rates of SARS-CoV-2 infection [20,21]. Thus, understanding the dynamics and drivers of under-vaccination and vaccine hesitancy in this workforce is important in reducing the negative impacts of COVID-19 and other emerging pathogens through future vaccination programs. 

The objective of this study was thus to understand the frequency and drivers of COVID-19 vaccination coverage and hesitancy within an agricultural community in rural Guatemala, in order to design an intervention to improve COVID-19 vaccination coverage (Appendix A).

## 2. Materials and Methods

We conducted a cross-sectional survey to evaluate knowledge, attitudes, and practices towards COVID-19 illness and freely available vaccines. This survey was embedded in the enrollment visit of a prospective cohort study to characterize asymptomatic and pre-symptomatic SARS-CoV-2 transmission among banana farmworkers’ households and workplaces. The study was conducted between 28 September 2021, and 11 April 2022, within two rural communities (Los Encuentros, Quetzaltenango, and Chiquirines, San Marcos) in southwest lowland Guatemala, approximately 50 km from the border of Chiapas, Mexico. The communities are monolingual Spanish-speaking. The population suffers from high rates of year-round food insecurity and child undernutrition, diarrheal disease, maternal depression, and maternal and child morbidity and mortality [22,23,24]. Previous surveys (2015, 2017–18) conducted among farm workers in the same communities found a predominantly young, male, and economically vulnerable workforce, in which farmworkers tend to be the primary income earners for their households; workers report high rates of food insecurity, poverty, and communicable diseases, as well as low access to healthcare [25]. Migration to the US from this community is frequent, including 3% in 2022 from a separate community cohort study (unpublished).

Inclusion criteria for the parent study and survey were the following: ≥1 member of the household employed in the agricultural sector; they must live in the selected communities; ≥75% of the people living in the household must consent to participate; and they must be eligible to receive COVID-19 vaccines at the time of the survey (age ≥ 12 years; survey inclusion criteria only). The questionnaire (S2: English and Spanish versions of the survey.) included an adapted Spanish-language COVID-19 “Vaccine Confidence” survey developed by the Centers for Disease Control and Prevention (CDC) [26]. The survey included the following topic areas and questions: demographics (age, sex, ethnicity); work environment (occupation, job site); pre-existing medical conditions (asthma, pulmonary disease, kidney disease, heart disease, diabetes, blood disorder, neurologic disease, and liver disease); COVID-19 knowledge, attitudes, and practices (KAP; prior exposure, level of concern, confirmatory testing, use of personal protective equipment (PPE)); COVID-19 vaccination history (date(s), vaccine type(s), number of doses); history of and reason(s) for vaccine refusal; and history of non-access to vaccines and reason(s). The survey also collected specific questions on COVID-19 vaccine hesitancy and decision making, including the following: perceived vaccination safety, preferences for vaccination (site), motivations for vaccination, peer decision making for COVID-19 vaccination, use of PPE if vaccinated, trust in public health institutions, quantify of information available on COVID-19 vaccination, and exposure to COVID-19 vaccine misinformation. The survey was administered verbally to adults and children (accompanied by adults) by trained study nurses at the participants’ homes or workplaces, and responses were recorded via the REDCap application (University of Colorado Anschutz Medical Campus, Aurora CO, USA) on a smartphone.

At the time of the survey, COVID-19 vaccination was offered only at Ministry of Health posts (usually two days/week) and through workplace vaccination programs by the Institute of Guatemala Social Security system (IGSS), such as the agribusiness that employed at least one member of each household. Vaccines available in Guatemala at the time of the survey included mRNA−1273 (mRNA, Moderna, Cambridge, MA, USA), BNT162 b2 (mRNA, Pfizer-BioNTech, New York, NY, USA), ChAdOx1-S (viral vector, AstraZeneca, Cambridge, England), and Gam-COVID-Vac (virus vector, Sputnik V, Moscow, Russia) All of these vaccines require two doses in the primary series. In Guatemala, individuals ≥ 12 years of age became eligible to receive COVID-19 vaccine in September 2021. Vaccination data were obtained directly from the national vaccination registry and verified by self-report of participants. 

Data analysis was conducted using SPSS^®^ software (version 25, Chicago, Il, USA). Participants with ≥1 dose of the COVID-19 vaccine were considered vaccinated. Descriptive statistics were used to characterize survey responses. The Mann–Whitney U test was used for median comparisons, and Pearson’s chi-square/Fischer’s exact tests were used for proportions; a *p*-value < 0.05 was considered statistically significant. The study was approved by the University of Colorado (COMIRB, protocol #21–2551), Universidad del Valle de Guatemala (UVG), and CDC ethics committees; it was funded by the CDC (CDCGH002243).

## 3. Results

From 28 September 2021, to 11 April 2022, we enrolled 340 individuals (86% of 394 eligible individuals) from 74 households (see map, Figure 1); overall, 233 individuals (69%) were ≥12 years old and 190 individuals (56%) were ≥18 years old (Table 1). Households and individuals were similar between the Chiquirines and los Encuentros communities in terms of demographics and exposure risks, though the mean monthly household income was slightly greater in Chiquirines (USD 250 vs. USD 197.37, *p* = 0.04). Overall, the median monthly household income was USD 210.53 (= quartile deviation (QD) = USD 92.11). Of the 340 enrolled subjects, 177 (52%) were female, 323 (95%) were of ladino/mestizo (mixed Spanish/indigenous) ethnicity, and of those >15 years, 107 (53%) worked outside the home. The median age was 21 years (QD = 12.25, range = 0–73 years). No children reported school attendance, as all schools in the community were closed because of the COVID-19 pandemic.

At the time of the survey, 4 (2%) respondents reported prior COVID-19 disease; 127 (55%) reported receiving >1 dose of a COVID-19 vaccine, which included mRNA−1273 (*n* = 106, 83%), ChAdOx1-S (*n* = 15, 12%), BNT162 b2 (*n* = 4, 3%), Gam-COVID-Vac (*n* = 1, 1%), and Cansino (*n* = 1, 1%; administered in nearby Mexico). Of those, 89 (70%) reported receiving two doses of COVID-19 vaccine, with the majority (*n* = 88, 97%) receiving mRNA−1273 vaccine. Only two subjects (2%) reported receiving a third dose of vaccine, which included mRNA vaccines. Only 11 participants (5%) reported having ever refused any vaccine in the past; common reasons for refusal of past vaccines were thinking a vaccine was unnecessary (*n* = 7, 64%), concern about side effects (*n* = 2, 18%), and someone else telling the respondent that the vaccine was unsafe (*n* = 2, 18%) (Table 2). 

Forty (17.2%) of the respondents reported previously being unable to obtain a routine vaccination despite intent. The most common reasons provided were the vaccine not being available (*n* = 14, 32%), not knowing where to get vaccinated (*n* = 7, 16%), not being able to leave work during clinic hours (*n* = 5, 12%), and not knowing where to get good and reliable information about the vaccine (*n* = 5, 12%).

In comparing vaccinated and unvaccinated individuals ≥ 12 years (eligible) at the time of the survey, vaccinated individuals were more likely to be male (59% vs. 27%, *p* < 0.01), older (median age = 33 vs. 25 years, *p* < 0.01), work outside of the home (69% vs. 24%, *p* < 0.01), or work on a farm (70% vs. 43%, *p* = 0.008) (Table 3, Figure 2). Of those vaccinated, four (3%) reported having refused another type of vaccination previously (prior to the COVID-19 pandemic) vs. seven (7%) of those who were unvaccinated (*p* = 0.23). There were no differences between vaccinated and unvaccinated individuals among reasons reported for being unable to access vaccines in the past.

Among those aged > 18 years (Table 3) at the time of the survey, vaccinated individuals were more likely to be moderately or very worried about COVID-19 (*n* = 36, 31%) compared to unvaccinated individuals (*n* = 13, 18%; *p* = 0.04). The most frequently reported motivations for COVID-19 vaccination are shown in Table 3 and include protecting the health of the participant, which differed between vaccinated and unvaccinated (24% vs. 11%, respectively; *p* < 0.01); there was no difference between motivation related to protecting their family/friends (68% vs. 73%, *p* = 0.53), protecting their community (3% vs. 3%, *p* = 0.94), going back to work or school (2.5% vs. 3, *p* = 0.94), or being encouraged by others to get vaccinated (2% vs. 0%, *p* = not calculable). Compared to vaccinated individuals, unvaccinated individuals were more likely to report little/no confidence in public health institutions (38% vs. 55%, *p* = 0.02). Among 73 (38%) unvaccinated participants > 18 years, 25% reported they would obtain the vaccine as soon as possible, 40% reported they would obtain it but would wait, 11% responded they would not obtain vaccine, and 25% reported they were unsure. 

## 4. Discussion

This cross-sectional survey shows that although vaccine refusal prior to the COVID-19 pandemic was rare (5%), in these two agricultural communities in Guatemala, nearly half (45%) of vaccine-eligible participants remained unvaccinated against COVID-19 (with any dose) one year following COVID-19 vaccine availability. This raises the question of how COVID-19 vaccine distribution and information may have differed in this community compared to previous vaccine programs, potentially contributing to lower uptake and confidence. 

The majority of respondents not yet vaccinated were female homemakers. Many of the adult males in these communities were employed at large agribusinesses that offered recommended COVID-19 vaccination through the workplace, suggesting that greater access to vaccine (through the workplace) may improve vaccination coverage. Greater vaccination rates and intention to vaccinate among males has been observed in other LMICs as well [27,28], though further information is needed on the reasons for greater vaccination rates in these other settings. Other studies from Latin America also found structural barriers as a common reason for non-vaccination against COVID-19, highlighting the interplay between vaccine access and hesitancy [29]. Extending workplace programs to family members of employees and implementing home- or community-based (places of worship, community gathering places) interventions may increase access for those who remain unvaccinated. Indeed, engaging religious leaders to reduce vaccine hesitancy has been proposed in Guatemala and globally [10,15,30].

We also found evidence of vaccine hesitancy. Unvaccinated individuals reported lower confidence in public health institutions, not receiving enough information about vaccines, and inability to find accurate and timely information about COVID-19 vaccines. Though data are limited from LMICs, similar results were observed among a qualitative study of migrant Latinx farm workers in California, USA, which found misinformation and lack of trust in institutions as primary themes that affect attitudes towards vaccination [31].

These data suggest a need for improved and more effective public health messaging in this community and similar rural areas. Indeed, this need is only compounded by the ongoing “infodemic” of misinformation from other sources, which has been associated with increased vaccine hesitancy [2,32,33,34]. It is possible that delays in achieving vaccine access in these rural communities may have served to undermine vaccine acceptance and “opened the door” to increased misinformation, contributing to greater hesitancy and decreased acceptance once the vaccine became more widely available. Future studies, which include a repeat of the vaccine hesitancy survey in the same household cohort, will help clarify this question and also address changing beliefs and behaviors surrounding COVID-19 vaccination. 

This study is limited in that it was carried out during the enrollment visit of a SARS-CoV-2 household transmission study, and it was restricted to households of workers employed at a large agribusiness, thus limiting generalizability. However, agricultural workers comprise 35% of the overall labor force in Guatemala and thus represent an important subpopulation in which to study vaccine access and hesitancy. This group is also essential to both local and global food security. Qualitative research interviews would allow a more comprehensive assessment of the complex drivers of vaccine hesitancy and barriers to access in these and similar communities. The survey was also carried out at a single timepoint and may not reflect changing vaccine attitudes, though we aim to address this limitation with follow-up surveys, as outlined above.

## 5. Conclusions

These findings provide an opportunity to implement improved evidence-based public health messaging and access strategies in the community. Building on the need to communicate at the household level and to increase community engagement and information sharing, we are designing a public health messaging strategy that involves a cadre of respected community leaders (members of the Community Development Council–COCODE, nurses from the health posts, midwives, teachers, and religious leaders, among others) who will be trained to provide COVID-19 vaccine information in the community. In parallel, health posts in the community are implementing a house-to-house vaccination program with public health nurses, which may reach the population that has not been vaccinated through their workplaces. After this process, this survey will be conducted again to identify changes in vaccine hesitancy and uptake.

## Figures and Tables

**Figure 1 vaccines-11-01059-f001:**
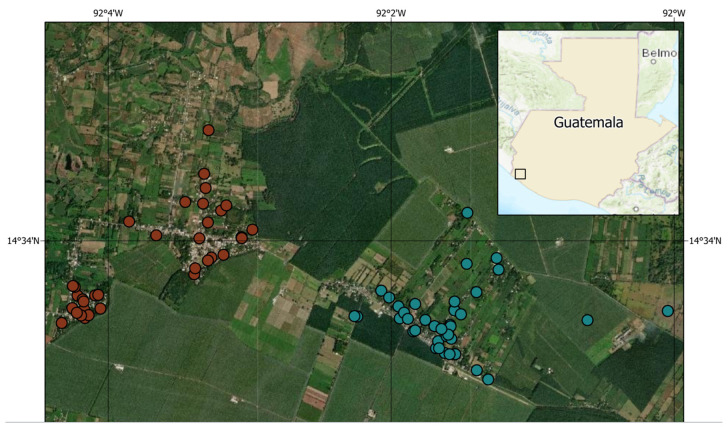
Map showing the study catchment areas and households from the Chiquirines and los Encuentros municipaities that were included in the vaccine hesitancy survey, 28 September 2021, to 11 April 2022.

**Figure 2 vaccines-11-01059-f002:**
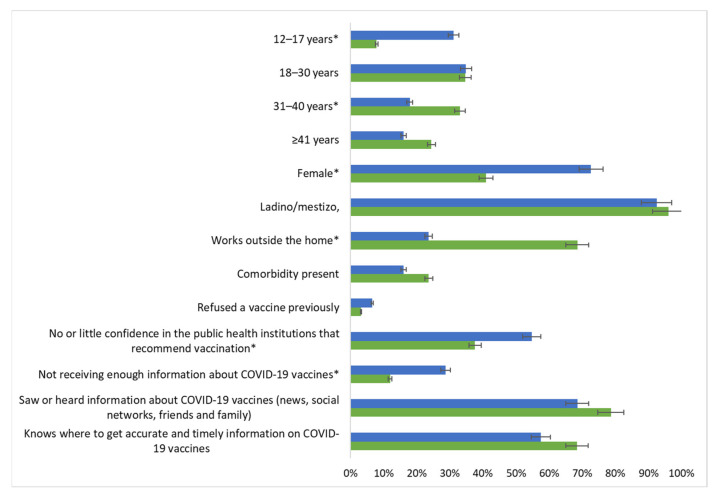
Comparison of community members vaccinated and not vaccinated against COVID-19 in the Trifinio Region of southwest Guatemala. * Considered statistically significant (*p*-value < 0.05).

**Table 1 vaccines-11-01059-t001:** Sociodemographic characteristics of enrolled households in two rural communities in the Trifinio Region of Southwest Guatemala, 2021–2022.

Variable	Los Encuentros (*n* = 40)	Chiquirines (*n* = 34)	*p*-Value ***
Household			
Persons living at home, median (QD)	5 (1)	5 (1)	0.76
Children living at home, median (QD)	2 (1)	3 (1)	0.22
Beds at home, median (QD)	3 (1)	3 (1.5)	0.90
Monthly household income, USD, median (QD) *	197.37 (65.8)	250 (72.37)	0.04
Individual	(*n* = 181)	(*n* = 159)	
Age, median (QD)	21 (6.5)n	20 (12.5)	0.93
Female, *n* (%)	95 (52)	82 (52)	0.86
Ethnicity			
Ladino/mestizo **, *n* (%)	166 (92)	157 (99)	<0.01
Indigenous	3 (2)	1 (0.5)	
Do not know	12 (6)	1 (0.5)	
Reports comorbidity, *n* (%)	33 (18)	20 (13)	0.15
Work outside the home, ≥15 years old, *n* (%)	53/112 (47)	54/90 (60)	0.29
COVID-19 vaccine, ≥12 years old, *n* (%)	67/123 (54)	60/110 (55)	0.99

Abbreviations: QD = quartile deviation, * Exchange rate: USD 1 (1 US dollar) = GTQ 7.6 (7.6 Guatemala quetzales). ** Mixed Spanish/indigenous descent. *** *p* value = the Mann–Whitney U test was used for median comparisons, and Pearson’s chi-square/Fischer’s exact tests were used for proportions; a *p*-value < 0.05 was considered statistically significant for all comparisons.

**Table 2 vaccines-11-01059-t002:** COVID-19 vaccination and vaccine hesitancy data from vaccine-eligible survey respondents.

COVID-19 Vaccination in Participants ≥ 12 Years Old.	*n* (233)
Received COVID-19 vaccine * *n* (%)	127 (55)
Received 1 dose of COVID-19 vaccine, *n* (%)	36 (28)
Received 2 doses of COVID-19 vaccine, *n* (%)	89 (70)
Received 3 doses (booster) of COVID-19 vaccine, *n* (%)	2 (2)
First dose COVID-19 (*n* = 127)	
Moderna, *n* (%)	106 (83)
AstraZeneca, *n* (%)	15 (12)
Pfizer-BioNTech, *n* (%)	4 (3)
Sputnik V, *n* (%)	1 (1)
Other (Cansino), *n* (%)	1 (1)
Second dose COVID-19 (*n* = 91)	
Moderna, *n* (%)	88 (97)
AstraZeneca, *n* (%)	2 (2)
Sputnik V, *n* (%)	1 (1)
Third dose (booster) COVID-19 (*n* = 2)	
Moderna, *n* (%)	1 (50)
Pfizer-BioTech, *n* (%)	1 (50)
Vaccine Hesitancy	
History of COVID-19 (self-report), *n* (%)	4 (2)
Refused a routine vaccine previously, *n* (%)	11 (5)
Reasons for vaccine refusal (*n* = 11) **	*n* (%)
I did not think it was necessary, *n* (%)	7 (63)
I am concerned about side effects, *n* (%)	2 (18)
Someone else told me vaccine was not safe, *n* (%)	2 (18)
Other, *n* (%)	3 (27)
Previously (pre-pandemic) wanted to receive a routine vaccine (any type) but was unable to do so, *n* (%)	40 (17)
Most common reasons for previously being unable to obtain a routine vaccine despite intent (*n* = 40)	*n* (%)
Vaccine not available at my health clinic or in my community	14 (32)
I did not know where to get vaccinated	7 (16)
I did not know where to get good and reliable information about the vaccine	5 (12)
Could not afford the vaccine	1 (2)
It is not possible to leave my work to receive the vaccine during clinic hours	5 (12)
Another reason	11 (26)

* A person with ≥1 dose of COVID-19 vaccine was considered as vaccinated. ** May choose more than one answer.

**Table 3 vaccines-11-01059-t003:** Comparison of participants ≥ 12 years old vaccinated and unvaccinated against COVID-19.

Variable	Vaccinated *n* = 127 (%)	Not Vaccinated *n* = 106 (%)	*p*-Value **
Age, median (QD)	33 (7)	25 (10)	<0.01
12–17	10 (8)	33 (31)	<0.01
18–30	44 (35)	37 (35)	0.96
31–40	42 (33)	19 (18)	<0.01
≥41	31 (24)	17 (16)	0.11
Female, *n* (%)	52 (41)	77 (73)	<0.01
Ladino/mestizo, *n* (%)	122 (96)	98 (93)	0.45
Works outside the home, *n* (%)	87 (69)	25 (24)	<0.01
Comorbidity present, *n* (%) *	30 (24)	17 (16)	0.15
Refused a vaccine previously, *n* (%)	4 (3)	7 (7)	0.23
Reasons for vaccine refusal (may select ≥ 1)	*n* = 4	*n* = 7	
did not think it was necessary	2 (50)	5 (71)	0.57
I did not know where to get reliable information	1 (25)	0 (0)	n/a
I was concerned about side effects	0 (0)	2 (29)	n/a
Someone else told me that the vaccine was not safe	0 (0)	2 (29)	n/a
Fear of needles	0 (0)	1 (14)	n/a
I was not able to leave my job/house to go to get vaccinated	1 (25)	0 (0)	n/a
Previously (pre-pandemic) wanted to obtain a vaccine (any type) but was unable to do so	19 (15)	21 (20)	0.32
Reasons why you were unable to be vaccinated (pre-pandemic, may select ≥ 1)	*n* = 19	*n* = 21	
Vaccine not available at my health clinic or in my community	5 (26)	9 (43)	0.27
I did not know where to get vaccinated	3 (16)	4 (19)	1.00
I did not know where to get good and reliable information about the vaccine	2 (11)	3 (14)	1.00
Could not afford the vaccine	1 (5)	0 (0)	n/a
It is not possible to leave my work to receive the vaccine during clinic hours	4 (21)	1 (5)	0.17
Another barrier to receiving the vaccine	4 (21)	7 (33)	0.48
Participants ≥ 18 years old	*n* = 117 (%)	*n* = 73 (%)	
How concerned are you or were you about contracting COVID-19?			
Not at all worried	35 (30)	33 (45)	0.32
Somewhat concerned	46 (39)	27 (37)	0.74
Moderately Concerned	14 (12)	7 (10)	0.61
Very concerned	22 (19)	6 (8)	0.04
Primary motivation to be vaccinated against COVID-19			
Protecting my health	28 (24)	8 (11)	0.02
Protect the health of my family and friends	80 (68)	53 (73)	0.53
Protecting the health of my community	3 (2.5)	2 (3)	0.94
Back to work or school	3 (2.5)	0 (0)	n/a
Because others encouraged me to get vaccinated.	2 (2)	0 (0)	n/a
Other	1 (1)	0 (0)	n/a
Not sure	0 (0)	10 (13)	n/a
No or little confidence in the public health institutions that recommend vaccination	44 (38)	40 (55)	0.02
Saw or heard information about COVID-19 vaccines (news, social networks, friends, and family)	92 (79)	50 (69)	0.25
Not receiving enough information about COVID-19 vaccines	14 (12)	21 (29)	0.01
Knows where to get accurate and timely information on COVID-19 vaccines	80 (68)	42 (57)	0.25

* Comorbidity refers to participants who reported at least one of the following medical conditions present: asthma, pulmonary disease, kidney disease, heart disease, diabetes, anemia, neurological disease, and liver disease. ** *p*-value: the Mann–Whitney U test was used for median comparisons, and Pearson’s chi-square/Fischer’s exact tests were used for proportions; a *p*-value < 0.05 was considered statistically significant for all comparisons.

## Data Availability

The data presented in this study are available on request from the corresponding author. Restrictions apply to the availability of some of the data due to participant confidentiality, and therefore it has not been made publicly available.

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
