# Peer review of "COVID-19 Attitudes and Vaccine Hesitancy among an Agricultural Community in Southwest Guatemala: A Cross-Sectional Survey"

_vaccines, 2023, doi:10.3390/vaccines11061059_

Round 1
Reviewer 1 Report
Several studies published during the past few years have investigated the reasons for vaccine hesitancy in particular populations. This report on a cross-sectional study conducted among an agricultural community is SW Guatemala is another of those well planned and conducted studies that sought to clarify the reasons for COVID-19 vaccine coverage and hesitancy. The paper reports a situation that is not unique to this study, that a large number of donated and purchased vaccine doses had expired presumably because of low accessibility and demand throughout the country. The object of this study was to design an intervention to improve COVID-19 vaccination coverage. The survey instrument appears to have been well designed and was appropriate to provide useful information on participants’ reasons for COVID-19 vaccine acceptance or hesitancy. The number of participants appears to be adequate to enable reliable statistical analysis and to provide useful information. When the survey was conducted the number of respondents reporting prior COVID-19 disease was low (2%) and only 11 participants (5%) reported having ever refused any vaccine while Table 2 reveals that 17% pre-pandemic wanted to receive a routine vaccination of any type but were unable to do so indicating an enduring accessibility problem! A previous study that comes to mind was conducted comparing vaccine hesitancy in two populations in Italy, one in Palermo in Sicily and the other in Bologna. I believe that in that study the influence of advice from older family members in Palermo probably was a cause of increased vaccine hesitancy re COVID-19 vaccine acceptance. In the present study many female homemakers in Guatemala had not yet been vaccinated however adult males, many working in large agribusinesses that offered COVID-19 workplace vaccinations, had indeed been vaccinated. As the authors point out, providing the vaccination benefits to other family members would certainly extend the benefits of COVID-19 vaccine delivery within the population. Table 3 reveals a worrying % of respondents who indicated having little or no confidence in public health institutions that recommend vaccination. Community education is clearly something that could be improved!
Generally I found this to be a well designed and conducted study that reveals reasons for vaccine acceptance or refusal in this population in Guatemala.
Author Response
We thank the reviewer for their feedback. There weren't any suggested changes, so we did not make any modifications to the manuscript. The study mentioned from Italy is interesting in that older family members in this other study impacted vaccine hesitancy on study participants. We unfortunately did not collect these data, but this is definitely something we will consider with future iterations of this survey (which are planned).
Reviewer 2 Report
The manuscript seems really interesting though in my opinion it seems more proper to be submitted as short communication or letter than a scientific article. The used language is excellent. I would suggest improvements especially in the discussion section, where it would be imperative to compare the results with other published results in papers and reviews. It would be highly required to present at least some of the results as graphics. In addition, readers who have never been or read about Guatemala don't know where are located the corresponding study communities. I also please to add a map to the manuscript in order to better illustrate this information. More comments will be found in the attached file.

Reviewer 3 Report
This paper is important as it addresses the critical issue of vaccine hesitancy and low vaccination rates in a rural Guatemalan community, which has broader implications for public health and COVID-19 containment in Latin America. By adopting a CDC questionnaire to evaluate access to and attitudes toward COVID-19 vaccines, the study sheds light on the gender and occupation disparities among vaccinated and unvaccinated individuals
Minor issues
1. In the introduction, write a short paragraph explaining the Health Care System of Guatemala so that a reader can understand the situation.
2. In the material and methods, the authors say that the questionnaire in Spanish is provided in the supplementary file. However, the supplementary file is not the Spanish version of the questionnaire. It is a translation into Spanish of the submitted manuscript. The authors should submit both Spanish and English questionnaire versions as Annex.
3. Table 1 is difficult to understand
Instead of writing q1 and q3, write the Quartile deviation. Quartile deviation, also known as the semi-interquartile range, is a measure of dispersion focusing on the spread of the middle 50% of the data. It is calculated as half the difference between the first quartile (Q1) and the third quartile (Q3). The formula for quartile deviation is Quartile Deviation (QD) = (Q3 - Q1) / 2. If used, make the table easy to read because in each column, there are always two numbers mean(SD) or median(QD)
The use of asterisks in the table is confusing. The authors wrote
*Q7.6 = $USD 1.
*Mixed Spanish/indigenous
. **p-value= the
That should be avoided. You can have two notes with the same symbol, e.g.,*, **, ***
This text is confusing because of the similarity with Q1, Q3,*Q7.6 = $USD 1. It could not be very clear,
It should be something like ** Quetzal/USD Change Q7.6 = $USD 1.
Round 2
Reviewer 2 Report
All the suggested modifications are reported in the manuscript and more details are found in the reply to the reviewer. I would suggest to the Editor in Chief to accept it in the present form
Author Response
We appreciate the reviewer's timely follow-up and their thorough review